# Circulating microRNAs as Potential Biomarkers in Pancreatic Cancer—Advances and Challenges

**DOI:** 10.3390/ijms241713340

**Published:** 2023-08-28

**Authors:** Attila A. Seyhan

**Affiliations:** 1Laboratory of Translational Oncology and Experimental Cancer Therapeutics, Warren Alpert Medical School, Brown University, Providence, RI 02912, USA; attila_seyhan@brown.edu; 2Department of Pathology and Laboratory Medicine, Warren Alpert Medical School, Brown University, Providence, RI 02912, USA; 3Joint Program in Cancer Biology, Lifespan Health System and Brown University, Providence, RI 02912, USA; 4Legorreta Cancer Center, Brown University, Providence, RI 02912, USA

**Keywords:** microRNAs, circulating miRNAs, post-transcriptional gene regulation, diagnostic, prognostic, predictive, biomarkers, liquid biopsy, pancreatic cancer

## Abstract

There is an urgent unmet need for robust and reliable biomarkers for early diagnosis, prognosis, and prediction of response to specific treatments of many aggressive and deadly cancers, such as pancreatic cancer, and liquid biopsy-based miRNA profiling has the potential for this. MiRNAs are a subset of non-coding RNAs that regulate the expression of a multitude of genes post-transcriptionally and thus are potential diagnostic, prognostic, and predictive biomarkers and have also emerged as potential therapeutics. Because miRNAs are involved in the post-transcriptional regulation of their target mRNAs via repressing gene expression, defects in miRNA biogenesis pathway and miRNA expression perturb the expression of a multitude of oncogenic or tumor-suppressive genes that are involved in the pathogenesis of various cancers. As such, numerous miRNAs have been identified to be downregulated or upregulated in many cancers, functioning as either oncomes or oncosuppressor miRs. Moreover, dysregulation of miRNA biogenesis pathways can also change miRNA expression and function in cancer. Profiling of dysregulated miRNAs in pancreatic cancer has been shown to correlate with disease diagnosis, indicate optimal treatment options and predict response to a specific therapy. Specific miRNA signatures can track the stages of pancreatic cancer and hold potential as diagnostic, prognostic, and predictive markers, as well as therapeutics such as miRNA mimics and miRNA inhibitors (antagomirs). Furthermore, identified specific miRNAs and genes they regulate in pancreatic cancer along with downstream pathways can be used as potential therapeutic targets. However, a limited understanding and validation of the specific roles of miRNAs, lack of tissue specificity, methodological, technical, or analytical reproducibility, harmonization of miRNA isolation and quantification methods, the use of standard operating procedures, and the availability of automated and standardized assays to improve reproducibility between independent studies limit bench-to-bedside translation of the miRNA biomarkers for clinical applications. Here I review recent findings on miRNAs in pancreatic cancer pathogenesis and their potential as diagnostic, prognostic, and predictive markers.

## 1. Introduction

Pancreatic ductal adenocarcinoma (PDAC) is the most common and highly aggressive and deadly pancreatic cancer characterized by an unparalleled mortality-to-incidence ratio and with a poor prognosis, and the number of patients with this disease is increasing [1,2,3]. This is partly due to the rapid course of disease and absence of symptoms at the initial stage of the disease as well as late detection and lack of effective biomarkers for monitoring its progress. The early and aggressive local invasion and high rate of metastasis are the main characteristics of pancreatic cancer with the worst rates of overall survival. The five-year survival rate has only just reached ~11% [4], and the standard of therapy improves survival in a range of months. In addition, approximately 50% of newly diagnosed pancreatic cancers are metastatic, with an average survival of patients less than one year [2]. It is estimated that 64,050 individuals will be diagnosed and 50,550 individuals will die from PDAC in the United States [5]. Due to the rising incidence of disease and lack of or limited effectiveness of treatments, pancreatic cancer is projected to become the second-leading cause of cancer-related deaths by 2030 [3,5]. Because surgery is considered to be the only curative treatment modality for pancreatic cancer, early detection has the potential to significantly improve outcomes [5].

Such stark statistics are driven are driven by the stealthy onset of the disease that often features non-specific symptoms and the absence of effective biomarkers for screening; the majority of pancreatic cancer cases (82%) are detected too late, and most diagnoses are made after the opportunity for surgical intervention has passed, resulting in surgically unresectable, locally advanced or metastatic cancer [5]. Even after complete surgical resection, 80% of patients experience recurrence, and at this stage the disease is almost universally deadly [6,7,8]. On the other hand, in the metastatic setting, the adjuvant standard of care first-line chemotherapy treatment options [i.e., FOLFIRINOX (5-fluorouracil, leucovorin, irinotecan, and oxaliplatin) and the combination of gemcitabine and nab-paclitaxel] have been shown to benefit and reduce recurrence risk (overall survival benefit of 19.4 months over gemcitabine alone, 54.4 months vs. 35.0 months) [5,7]. It has been shown that FOLFIRINOX improves the median overall survival by 4.3 months (11.1 months vs. 6.8 months) as compared to gemcitabine monotherapy [7,9]. Likewise, the combination of nab-paclitaxel with gemcitabine was shown to improve median overall survival by 1.8 months (8.5 months vs. 6.7 months) [10].

Although screening high-risk individuals (e.g., those with predisposing genetic conditions with a lifetime pediatric cancer risk of >5–15%, high-risk pancreatic lesions, and strong family history, obesity, and type 2 diabetes) [2] with magnetic resonance imaging (MRI) or endoscopic ultrasound (EUS) [11] is recommended, the optimal timing of screening and the benefit of the combination of these tests remains to be determined. Likewise, screening tests to ascertain response to treatment and disease recurrence are also limited to MRI imaging and infrequently tumor markers measured in serum that suffer from poor sensitivity and specificity [12,13]. Importantly, while screening high-risk individuals might have some benefits, it has been shown that universal screening of the general public is ineffective [11].

Because of this, there is an urgent need for the discovery and development of easily detectable biomarkers as diagnostic tools to detect disease at earlier stages, guide therapy selection, monitor treatment response, and predict the recurrence of the disease and hence improve the survival rates of patients [14].

Cancer is a complex and heterogeneous disease that evolves through successive genetic and genomic as well as epigenetic changes that support tumorigenesis. It is a disease that is driven by both internal stimuli such as genome alterations leading to germline and somatic alterations and external stimuli that play a role in the introduction of some genome alterations and expression patterns of certain genes and cellular signaling pathways associated with cell growth, migration, and metastasis. Changes in the genome that affect gene function often result from genomic alterations including chromosomal translocations, insertions or deletions, amplifications, and single-nucleotide mutations or from the epigenome leading to activation of oncogenes and suppression of tumor suppressor genes [15]. As described in a recent review [16], the hallmarks of cancer make it even more challenging to tackle this complex disease.

Although the majority of oncology research is still focused on the dynamic variations of proteins and protein-coding RNAs (their corresponding coding sequences only account for ∼2% of the genome (https://www.genomicseducation.hee.nhs.uk/genotes/knowledge-hub/non-coding-dna/ (accessed on 26 July 2023)) [17,18,19]), the role of non-coding RNAs (ncRNAs) transcribed from the remaining 98% of the genome including microRNAs (miRNAs) play key roles in a multitude of biological processes in normal physiological states [20] but also in the development of various types of diseases including cancer [20], underscoring the importance of miRNAs and other ncRNAs in tumor initiation and progression. As such, miRNAs and their altered expression have been recognized as an additional molecular mechanism responsible for the pathological processes of many diseases [21,22,23] including innate immunity [24], autoimmunity and autoimmune diseases [25], viral infections [26,27,28,29], acute hepatitis [30], depression [31], anxiety [32], Alzheimer’s disease [33], Huntington’s disease [34], obesity, metabolic and cardiovascular diseases [35,36,37,38], diabetes [39,40,41,42,43,44], and many cancers [5,14,45,46,47,48,49,50,51,52,53,54,55,56,57,58,59,60,61,62,63,64,65,66,67,68,69,70,71,72,73], consequently, these miRNAs can be used to indicate the presence of a pathology and even the stage, progression, or genetic link of the disease. Supporting this, recent findings on miRNAs highlight their roles during tumor pathogenesis as well as in response to various therapies.

Although genetic, genomic, and epigenetic changes affect genome function, dysfunctions of various types of regulators are the primary molecular mechanisms responsible for the pathological processes of many cancers, miRNAs and their altered expression and members of the miRNA biogenesis pathway have also been recognized as additional molecular mechanisms implicated in the pathological processes of many cancers and have received great attention in recent years. To support this, new data on the molecular mechanisms responsible for the dysregulation of miRNA biogenesis and expression in cancer are continuously emerging. For example, it has been shown that genetic deletion or amplification, epigenetic methylation of miRNA genomic loci, and alterations affecting the transcription factor-mediated regulation of primary miRNA (pri-miRNA) as well as components involved in the miRNA biogenesis pathway frequently alter miRNA expression and function in many cancers. In addition, recent data indicate that other factors including oncogenic drivers such as mutations in the KRAS gene can also affect global miRNA biogenesis and effector function, leading to global miRNA dysregulation [74]. Because of this, miRNAs and their dysregulation have gained attention both from academia and industry as a research discipline both for understanding disease biology [59,75,76,77] and to develop them as potential diagnostic, prognostic, and predictive biomarkers as well as drug targets or as therapeutics [47,48,51,59,67,78].

Here, I review how miRNA expression is dysregulated in cancer with a focus on pancreatic cancer and discuss the clinical applications of miRNAs as potential biomarkers that may be used to facilitate patient diagnosis, prognosis, monitoring, and treatment in the oncology field.

## 2. microRNAs

After the discovery of the first microRNA (miRNA), lin-4, in 1993 in *Caenorhabditis elegans* [79,80], it was quickly recognized that microRNAs have been detected throughout the animal and plant kingdoms and some were shown to be highly conserved across species [81,82,83], with a conserved mechanism and broad functional significance.

MiRNAs are small non-coding RNAs, with an average of 22 nucleotides in length which are highly conserved and are naturally encoded in the genomes of various species [81,82,83,84] and play key functional roles in the regulation of gene expression at the transcriptional [85,86,87] and post-transcriptional [41,88,89,90,91] levels of their target mRNAs [41,89] including the regulation of cell function by regulating gene expression via the modulation of the stability and translation of mRNA [92] in a broad range of biological processes within cells and organisms, consequently affecting cell differentiation, cell proliferation, angiogenesis, and apoptosis, etc. [48]. In addition, miRNAs exhibit tissue-specific [67,93] and developmental expression patterns [75,76,77].

Emerging data suggest that new miRNAs are still being discovered [94] and curreently, it is estimated that there are >2588 human mature miRNAs in human cells with time- and tissue-dependent expression patterns. Of the >2588 miRNAs, 1115 are currently annotated in miRBase V22 [15,16,21,95,96,97,98]. It is estimated that these >2588 miRNAs regulate over 60% of the expression of human genes, demonstrating that they have important regulatory roles in diverse physiological and developmental processes as well as in the pathogenesis of various human diseases and disorders [21,22,23,24,25,26,27,28,29,30,31,32,33,34,35,36,37,38,39,40,41,42,43,44], including many cancers [5,14,45,46,47,48,49,50,51,52,53,54,55,56,57,58,59,60,61,62,63,64,65,66,67,68,69,70,71,72,73].

As illustrated in Figure 1, most miRNAs are transcribed from DNA sequences into nascent primary miRNAs (pri-miRNAs) which are then initially processed by DROSHA in the nucleus to generate precursor miRNA (pre-miRNA) [41,99]. As many as 40% of miRNA genes may lie in the introns or even exons of other genes [100]. Pre-miRNAs are subsequently exported from the nucleus to the cytoplasm by exportin 5 (XPO5) where they are further processed by DICER, producing small RNA duplexes with 2 nt. 3′ overhangs. These small double-stranded RNA duplexes with distinct 2 nucleotides 3′ overhangs are then loaded onto the Argonaute (AGO) protein, which retains only one strand of mature miRNA by removing the other strand [89]. The complex formed by the AGO and miRNA associates with cofactors such as GW182 (i.e., TNRC6A), and forms the effector complex, the RNA-induced silencing complex (RISC) [91]. The miRNA-RISC complex is responsible for the degradation of mRNA transcripts and translational suppression through interaction with the complementary mRNA target sequences predominantly located within the 3′-untranslated region (3′-UTR) of mRNAs (Figure 1) [101,102,103,104].

As discussed in the literature [59], a specific miRNA may target many different mRNAs [105]; while a particular messenger RNA may be targeted by several miRNAs, either simultaneously or in a context-dependent fashion [106], resulting in the cooperative repression effect [107,108].

## 3. miRNA Dysregulation in Cancer

Cancer is a complex and heterogeneous disease that evolves through successive genetic and genomic changes that support tumorigenesis [109]. Changes in the genome that affect gene function often result from genomic alterations such as chromosomal translocations, insertions or deletions, amplifications, and single-nucleotide mutations in the epigenome. These genetic and epigenetic alterations often trigger the activation of oncogenes and suppression of tumor suppressor genes [110]. To add to this picture, miRNAs also play an important role in organismal development, normal physiology, and disease state, including many hallmarks of various cancer types (Figure 2).

Although genetic, genomic, and epigenetic changes affecting genome function as well as the dysfunctions of various types of regulators are the primary molecular mechanisms responsible for the pathological processes of a multitude of human diseases, including many cancers, among which, miRNAs and their altered expression, and alterations and mutations in members of miRNA biogenesis pathways have also been recognized as additional molecular mechanisms implicated in the pathological processes of many cancers and have received great attention in recent years. Numerous studies have demonstrated the involvement of miRNAs in cancer pathogenesis by regulating the expression of their target mRNAs contributing to tumor growth, invasion, angiogenesis, and immune system evasion [111,112]. Furthermore, studies have shown that tumor-specific miRNA signatures can define cancer subtypes, patient survival, and treatment response [46,113,114], and notably, cancer-associated miRNAs can be detected in biological fluids, allowing less-invasive monitoring [115].

It is widely accepted that genetic deletion or amplification, epigenetic methylation of miRNA genomic loci, and alterations affecting the transcription factor-mediated regulation of primary miRNA (pri-miRNA) as well as factors involved in the miRNA biogenesis pathway frequently alter miRNA expression and function in many cancers.

Alterations throughout miRNA biogenesis can affect the availability of target mRNA, including many mRNAs associated with cancer development (Figure 3). As such, altered or dysregulated miRNAs or miRNA-processing machinery have been linked to many pathological processes, resulting in the failure to maintain the normal homeostatic state and eventually leading to malignant transformation, including many cancers [5,14,45,46,47,48,49,50,51,52,53,54,55,56,57,58,59,60,61,62,63,64,65,66,67,68,69,70,71,72,73].

As illustrated in Figure 3, various genetic or epigenetic alterations in key factors involving the miRNA biogenesis process can affect the availability of miRNA targets, including many targets associated with cancer development, as well as several modulators and proteins (e.g., EGFR), transcription factors (e.g., ETS1/ELK1, MYC), epigenetic regulators (e.g., demethylating protein KDM6A, a known tumor suppressor), the tumor suppressor gene TAP63, and miRNAs (e.g., miR-103) [116] that may interact with factors involved in miRNA biogenesis machinery. It has been shown that both KRAS and EGFR are essential mediators of pancreatic cancer development and they were shown to interact with Argonaute 2 (AGO2) to perturb its function [117].

Recent findings indicate that different types of miRNAs have been shown to regulate many of the hallmarks of cancer Figure 2 [71], either as a tumor suppressor (i.e., oncosuppressor miRs) or as oncogenes (i.e., oncomirs) [118,119,120,121,122,123,124,125,126,127,128]. MiRNAs play a critical role in the regulation of the expression of a multitude of genes, such as the cellular responses to environmental stresses including DNA damage, hypoxia, oxidative stress, and starvation.

Supporting this, miRNAs with an oncogenic function as well as tumor-suppressing function have recently been identified in various neoplastic malignancies, and dysregulation of miRNA expression is closely associated with cancer initiation, progression, and metastasis and shown to be associated with the origin, progression, therapeutic response, and patient survival of the disease [47,51,55,62,67,72,73,129,130].

For example, the tissue specificity of miRNAs [67,93], which is required for maintaining normal cell and tissue homeostasis [46], allows them to be used as potential biomarkers in diagnosing cancer of unknown primary origin [97,131].

Supporting these findings, recurrent genetic and epigenetic alterations of individual miRNAs and components involved in the miRNA-processing machinery and biogenesis pathway have been identified in many cancer types, and emerging data from several functional studies provide evidence for the mechanism of action of many potential oncogenic and tumor-suppressor miRNAs [118,119,120,121,122,123,124,125,126,127,128] with potential clinical applications as potential diagnostics and therapeutics.

## 4. Oncogenic Mutations in Key Factors Involved in the miRNA Biogenesis

As discussed in the literature [74], oncogenic mutations in factors involved in miRNA biosynthesis, processing, and export machinery are associated with many cancers. The majority of these oncogenic mutations in the microprocessing complex and export machinery result in a decrease in miRNA biosynthesis, leading to global dysregulation of miRNAs and the mRNA transcripts they regulate.

For example, mutations (e.g., E518K, rs417309 G/A, p.R32fs, copy-number loss, p.S92fsa) in DGCR8, a member of the nuclear microprocessing complex which binds pri-miRNAs to facilitate processing by DROSHA has been linked to various human cancers including Wilms tumor, thyroid carcinoma, FMGS, laryngeal cancer, breast cancer, and pineoblastoma [132,133,134,135,136].

Similarly, mutations (rs640831 C/A, rs1110386 G/Aa, rs486732 C/Aa, p.E500, p.R277C, p.Q136a) in RNASEN/DROSHA, a member of the nuclear microprocessing complex and a ribonuclease that cleaves pir-miRNA to pre-miRNAs have been linked to several human cancers including lung adenocarcinoma, ovarian cancer, bladder cancer, breast cancer, thyroid carcinoma, and pineoblastoma [136,137,138,139,140,141].

Mutations (p.S625C, p.P89A) in DHX9, a member of the nuclear microprocessing complex and a helicase that modulates nuclear miRNA processing have been linked to breast cancer [142].

Mutations (p.T1182del, rs2257082 A/G) in XPO5, which facilitates pre-miRNA export from the nucleus to the cytoplasm, have been linked to colon cancer, gastric cancer, endometrial cancer, and gastric cancer [143,144].

Likewise, mutations (germline: p.Y1180, p.Y819H, p.D1713Aa, p.R187a, p.G803Ra, p.E503Xa and somatic: p.A872Ta, p.E428Ka, p.E813Qa, p.D1709Na, p.D1709G, p.D1810Y, p.E1813Qa, p.Y1701) in DICER1, a member of the RISC complex, and an endoribonuclease that cleaves the stem-loop from pre-miRNAs, leading to the formation of mature miRNAs, have been linked to fetal lung adenocarcinoma, Sertoli–Leydig ovarian tumor, FMGS, hepatocellular carcinoma, Wilms tumor, pleuropulmonary blastoma, embryonal ovarian cancer, yolk-sac tumor, juvenile granulosa-cell tumor, and teratoma, pineoblastoma [136,145,146,147,148,149,150,151,152].

Mutations (p.P144fs, p.R353fs, p.M145fs, copy-number gain) in TARBP1, a member of the RISC complex and a binding partner of DICER that facilitates RISC assembly, have been linked to colon cancer, Wilms tumor, gastric cancer, and adrenal cancer [63,153,154,155].

Mutations (p.P127L) in PRKRA, a member of the RISC complex and a binding partner of DICER that facilitates RISC assembly, have been linked to ovarian cancer [63,153,154,155].

Mutations (copy-number gain) in AGO2, a member of the RISC complex and an endonuclease that has mRNA cleavage function, have been linked to head and neck cancer, multiple myeloma, breast cancer, and ovarian cancer [156,157,158].

Mutations (p.P115-Q118del, p.R1183fs, p.Trp804fs) in TNRC6A, a member of the RISC complex and a binding partner of AGO2 required for mRNA silencing, have been linked to esophageal cancer, gastric cancer, and colorectal cancer [63,159,160].

As an additional mechanism that promotes malignant transformation, recent data [74] show that mutant KRAS is also involved in the modulation of the activity of many members of miRNA processing and regulatory pathways. Supporting this, previous studies have shown that miRNA biogenesis and global miRNA expression levels are significantly dysregulated in cancers harboring KRAS mutations [161,162,163,164].

The gene encoding KRAS is the most mutated oncogene in many cancers including pancreatic ductal adenocarcinoma (PDAC), non-small-cell lung cancer (NSCLC), and colorectal cancer (CRC) [165,166,167,168]. Activating mutations in the KRAS gene result in the constitutive activation of downstream signaling cascades, consequently leading to constitutive and sustained proliferation, self-renewal, and increased vascularization [166,169,170,171,172]. Notably, the functional effects of KRAS mutations correlate with the oncogenic effects of global miRNA dysregulation [173].

Similar to the downstream effects of mutations in the key factors involved in miRNA biogenesis and processing pathways, oncogenic KRAS affects global miRNA dysregulation. This is achieved by directly regulating the activity of key factors involved in miRNA biogenesis, processing, and regulatory pathways by oncogenic KRAS as an additional mechanism to promote malignant transformation. Thus, this regulatory mechanism disrupted by oncogenic KRAS may represent a novel vulnerability in KRAS-dependent cancers such as pancreatic cancer and others.

Furthermore, data show that mutations in the factors involved in the miRNA biogenesis, processing, and export machinery have been linked to a decrease in miRNA biosynthesis, thus leading to the global dysregulation of miRNAs and the mRNA transcripts they target [74].

Beyond the effect of mutant KRAS on miRNA biogenesis and processing pathways, mutant KRAS also alters how miRNAs regulate their target RNA [74]. For instance, the mutant KRAS/AGO2 interaction was shown to dysregulate the effector function of miRNAs, as AGO2 and the miRNA/RISC are essential for miRNA localization and silencing of miRNA targets. Supporting this, cancer cells with KRAS mutations have been shown to have increased levels of AGO2 phosphorylation at serine 387 (AGO2S387) [174].

Beyond directly regulating members of the miRNA regulatory pathway, mutant KRAS was also shown to alter the coordination and the storage of miRNA-targeted transcripts in intracellular condensates involved in transcript regulation, such as stress granules (SGs) and processing bodies (PBs) [175,176,177,178].

Thus, a better understanding of the global miRNA dysregulation induced by oncogenic KRAS may reveal novel targetable pathways for therapeutic intervention.

Recently, two new drugs, sotorasib (Lumakras) [179] and adagrasib (Krazati) [180], have been approved to treat people with non-small cell lung cancer that has the KRAS G12C mutation, and those in development include MRTX1133 targeting KRAS G12D mutations. As such, concurrent targeting of various forms of mutant KRAS and individual members of the miRNA biogenesis and processing machinery might represent a synergistic therapeutic strategy to treat several types of cancer with KRAS mutation [181].

## 5. Liquid Biopsy in Pancreatic Cancer

Because pancreatic cancer is a highly aggressive and one of the most lethal types of cancer with a five-year survival rate remaining low (~11%), it is always diagnosed at an advanced stage and is resistant to therapy. Thus, there is an urgent unmet need for a reliable and robust biomarker that can detect asymptomatic premalignant or early malignant tumors and predict the response to treatment for pancreatic cancer patients [5].

Although used extensively for early diagnosis of cancers, carcinoembryonic antigen (CEA) [13] and carbohydrate antigen 19-9 (CA19-9) [13,182] have limitations in clinical practice and a single biopsy of tumor specimens is often not informative enough about tumor progression and metastasis [183]. As such, there is an unmet need for biomarkers that can detect disease in early stages and also predict the response to treatment and recurrence of tumors.

Because of this, it is fundamentally important for the detection of early asymptomatic disease, such that treatments can begin before the disease becomes unmanageable and metastatic and this can be only possible with the help of biomarkers that can be easily measured in real-time and repeatedly sampled to help diagnose disease in early stages and also help with the prediction of response to a specific treatment.

The liquid biopsy in pancreatic cancer, as discussed in detail in the literature [5,184] and illustrated in Figure 4, is a non-invasive or minimally invasive process followed by miRNA profiling that can inform the miRNA signatures for each type of cancer as well as stages of disease.

Unlike other methods such as invasive biopsy, liquid biopsy is a minimally invasive or non-invasive procedure depending on the bodily fluid collected (e.g., blood vs. saliva) for biomarker assessment. It gives a real-time and longitudinal assessment of the disease course and enables monitoring of treatment response. Importantly, liquid biopsy can be used to profile circulating miRNAs as biomarkers in real-time and longitudinally, which can inform the miRNA signatures for each type of cancer as well as stages of disease.

For example, as illustrated in Figure 4, in recent years, many circulating miRNAs identified as potential diagnostic and prognostic biomarkers in pancreatic cancer have been reported in the literature [66,185,186,187,188,189].

## 6. miRNAs as Potential Biomarkers

As discussed in the literature [190], dysregulated expression of miRNAs has been implicated in various human diseases and disorders [21,22,23,24,25,26,27,28,29,30,31,32,33,34,35,36,37,38,39,40,41,42,43,44], including many cancers [5,14,45,46,47,48,49,50,51,52,53,54,55,56,57,58,59,60,61,62,63,64,65,66,67,68,69,70,71,72,73], these miRNAs have been investigated in several clinical trials as indicators of the presence of a pathology and even of the stage, progression, or genetic link of disease. Because of this, miRNAs, especially those in circulation have been exploited as novel, minimally invasive, or non-invasive biomarkers because of their relatively high stability in circulation and other biofluids, sensitivity, and ease of detection using various detection methods in various biological fluids.

Notably, in some situations, one miRNA biomarker may be sufficient to identify a health outcome; however, in many other cases, a panel of well-defined and validated miRNAs has been used for increased diagnostic sensitivity and specificity as biomarkers [190].

Recently, many miRNAs were detected in cell-free conditions in circulation, including biofluids such as whole blood, plasma, serum, saliva, tears, urine, stool, pancreatic juice, breast milk, colostrum, seminal fluid, amniotic fluid, bronchial lavage, cerebrospinal fluid, peritoneal and pleural fluids, and other body-fluid types [191] and they exhibit specific expression patterns that are associated with altered physiological conditions and disease states [37,38,41,42,43,44,51,192,193,194].

miRNAs can be secreted into circulation in extracellular fluids and shuttled to target cells via extracellular vesicles, including exosomes, microvesicles, and apoptotic bodies under various physiologic and pathologic conditions, enabling them to function as chemical messengers to mediate cell–cell communication [193,195,196], or by binding to proteins such as Argonautes (AGO), especially AGO2 [195,197]. In addition, secreted miRNAs trafficked between different subcellular compartments have been shown to regulate translation, and even transcription [198].

The source of miRNAs, how accessible it is as a biomarker, how the miRNA is extracted from the biological specimen, and even how it is analyzed can significantly influence the miRNA analysis results. In biofluids [191], the miRNA content can originate through a few pathways, including extracellular vesicles (exosomes) [199,200,201,202]. Studies are investigating whether the enrichment of exosomes from biofluids or direct cell-free preparations are more reliable in miRNA isolation and measurements and some studies indicate that it may depend on the specific human disease [190].

It is important to note that exosomal vesicle purification can lower the RNA yield and integrity which can lead to more interindividual variability. Furthermore, a disease-specific alteration in exosome release and clearance, or administration of therapeutics can alter blood volume and can skew RNA yield and signature [203].

In addition to being contained within exosomes, significant portions of miRNAs appear to be contained in various cell types such as tumor cells [204], stem cells [205], macrophages [206], and adipocytes [207], all of which release exosomes into circulation with specific miRNA (exomiRs) content.

## 7. Circulating miRNAs Detected in Different Body Fluids

An ideal biomarker should be accessible via non-invasive techniques, be reasonably sensitive to distinguish early tumor presence before clinical symptoms and should be undetectable or low in healthy people [116,208].

Their non-invasive or minimally invasive accessibility, relative stability, low complexity, and ease of detection via various techniques make circulating miRNAs ideal candidates as biomarkers for several diseases [116].

Although miRNA profiling of tumor tissue has been used as a prognostic test, tumor sampling requires invasive techniques such as biopsy, aspiration biopsy, or excision surgery of prevailing tumors [208,209].

Data indicate that miRNAs can be detected in various body fluids including whole blood, plasma, serum, saliva, tears, urine, stool, pancreatic juice, breast milk, colostrum, seminal fluid, amniotic fluid, bronchial lavage, cerebrospinal fluid, peritoneal and pleural fluids, and others and that these miRNAs relate to different pathophysiological conditions [191].

It is reasonable to argue that circulating miRNAs detected in the above body fluids could be originating from lysed or dying cells [210]; however, data indicate that miRNAs are actively exported from the cells within exosomes [211] or as exosome- or microvesicle-free miRNAs attached to Argonaute2 complexes [212].

Although circulating miRNAs from plasma and serum are the most common method for profiling circulating miRNAs as biomarkers, other biofluids such as urine and saliva are also used as the source for circulating miRNAs [130].

As discussed in the literature [130], a recent report has analyzed miRNA expression data from 40 human healthy tissues and those from human body fluids, including serum, plasma, urine, bile, and feces, and shown a positive correlation between body fluid miRNAs and tissue miRNAs [213].

For example, miR-186-5p was shown to be overexpressed not only in tumor tissues and blood but also in the urine of bladder cancer patients [214].

Similarly, several miRNAs, including miR-210-3p, were shown to be elevated in the urine of patients with transitional cell carcinoma [215].

Another example is members of the let-7 family miRNAs. Data show that all members of let-7 families of miRNAs were significantly upregulated in urine from clear-cell renal cell cancer (ccRCC) patients [216] and a combination of urinary miRNAs including miR-34b-5p, miR-126-3p, and miR-449a-3p could be used for ccRCC diagnosis [217].

A panel of four miRNAs including miR-21-5p, miR-125b-5p, miR-155-5p, and miR-451-5p was found to be elevated in urine samples of breast cancer patients and shown their diagnostic utility [218].

miR-143-3p and miR-30e-5p were found to be elevated in urine samples of pancreatic ductal adenocarcinoma (PDAC) patients and were shown their potential utility as diagnostic biomarkers in the early stage of the disease [219].

Collectively, circulating miRNAs can be found in various bodily fluids and be obtained with minimally invasive or non-invasive methods, and potentially be used as biomarkers for human diseases including cancer.

In addition, miRNAs are relatively stable in those different types of body fluids making them ideal biomarkers for early diagnosis, prognosis, prediction, and monitoring of various types of cancers.

## 8. Circulating miRNAs as Potential Biomarkers in Pancreatic Cancer

Concerning their utility as potential biomarkers in pancreatic cancer, several studies have reported that the expression of several circulating miRNAs is altered in pancreatic cancer [14,62,66,67,129,130,185,186,187,188,189,220,221,222,223,224].

Table 1 illustrates examples of recent clinical findings on circulating miRNAs as prognostic markers and possible mechanisms of action and survival status in pancreatic cancer that have been reported in the literature [5,14,62,67,129,225,226].

Examples include miR-10b, which was shown to be upregulated in pancreatic cancer and shown to silence TIP30 and is linked to poor survival [227,228].

miR-21 which was shown to be upregulated in pancreatic cancer and silences PTEN, PDCD4, IL-6R, and CDK6 mRNAs is associated with worse survival [229,230,231].

On the other hand, miR-34a which was shown to be upregulated in pancreatic cancer and silences NOTCH, BCL2, and CDK6 mRNAs, is associated with better survival [209,234,235].

Likewise, the miR−200 family, which was shown to be downregulated in pancreatic cancer and inhibits E-cadherin, ZEBR, is associated with better survival [234,246,247].

Similarly, miR-873, which was shown to be downregulated in pancreatic cancer and inhibits PLEK2 mRNA via the pleckstrin-2-dependent PI3K/AKT pathway, is associated with better survival [253].

The let-7 family, which was shown to be downregulated in pancreatic cancer and inhibits KRAS, HRAS, and TRIM71, is associated with poor survival [243,244].

miR−216, which was shown to be downregulated in pancreatic cancer and inhibits ROCK1, is associated with poor survival [248,249].

miR-181a, which was shown to be upregulated in pancreatic cancer and inhibits RKIP mRNA and induces epithelial–mesenchymal transition such as cancer stem cell phenotype, is associated with poor survival [250].

miR-223, which was shown to be upregulated in pancreatic cancer and inhibits ZIC1 mRNA via PI3K/Akt/mTOR signaling pathway, is associated with poor survival [251].

miR-23b-3p, which was shown to be upregulated in pancreatic cancer and inhibits PTEN mRNA via the JAK/PI3K and Akt/NF-kappa B signaling pathways, is associated with poor survival [252].

miR-216a, which was shown to be downregulated in pancreatic cancer and inhibits WT1 mRNA and regulates KRT7 transcription, is associated with poor survival [254].

miR-139, which was shown to be downregulated in pancreatic cancer and inhibits RalB mRNA via the Ral/RAC/PI3K pathway, is associated with poor survival [255].

miR-1252-5p, which was shown to be downregulated in pancreatic cancer and inhibits NEDD9 mRNA. Importanly, miR-1252-5p is regulated by Myb, regulates NEDD9 mRNA inhibition, and is associated with poor survival [256].

miR-125b, which was shown to be downregulated in pancreatic cancer and inhibits NEDD9 mRNA via PI3K/AKT signaling pathway, is associated with poor survival [257].

miR-382, which was shown to be downregulated in pancreatic cancer and inhibits Anxa3 mRNA via PI3K/Akt signaling pathway, is associated with poor survival [258].

miR-519, which was shown to be downregulated in pancreatic cancer and inhibits PD-L1 mRNA, is associated with poor survival [259].

miR−155, which was shown to be upregulated in pancreatic cancer and inhibits TP53INP expression, is associated with poor survival [237,238].

Notably, miR-155-5p was upregulated in tumor tissues and plasma in pancreatic ductal adenocarcinoma, and the expressions in tissues were associated with tumor stage and poor prognosis [224,260,261].

It was also shown that long-term administration of gemcitabine in tumor cells led to the overexpression of miR-155-5p, inducing gemcitabine resistance via anti-apoptotic activity and that the elevated miR-155-5p expression correlated with chemoresistance and poor prognosis for pancreatic ductal adenocarcinoma patients receiving gemcitabine treatment [238].

Likewise, serum miR-373-3p was found to be downregulated in pancreatic cancer, and the miR-373-3p level was negatively correlated with tumor, node, and metastasis (TNM) stage, lymph node metastasis, and distant metastasis [262]. Furthermore, pancreatic cancer patients with reduced serum miR-373 levels exhibited shorter five-year overall survival [262].

Because individual miRNAs have less discriminatory power, a combination of miRNAs, such as miR-16 and miR-196 with CA19-9 as a combination biomarker panel, was shown to yield more robust results, as did a panel of seven miRNAs (i.e., miR-20a, miR-21, miR-24, miR-25, miR-99a, miR-185 and miR-191) which had already been shown to discriminate between PDAC and healthy sera [221]. On the other hand, a three-microRNA combination including miR-106b, miR-126, and miR-486 resulted in a slightly less accurate diagnosis [221].

Another study showed that changes in the levels of miR-25, GDF-15, and CA19-9 made accurate discrimination possible and that the plasma levels of the six miRNAs and MIC-1, CA19-9 were elevated in pancreatic cancer patients compared with those of healthy controls [263]. Among them, miR-20a, miR-21, miR-25, MIC-1, and CA19-9 could distinguish pancreatic cancer patients from those with other GI cancers or biliopancreatic diversion cancers [263]. These findings suggest that a combination of biomarkers has better diagnostic value compared with using a single marker.

Table 2 shows current miRNA human clinical trials as diagnostic biomarkers as of the data cutoff date of 30 June 2023.

Collectively, circulating miRNAs are potentially effective minimally invasive or non-invasive cancer biomarkers for clinical applications including cancer screening at the early stage, subtype classification, and predicting drug sensitivity for treatment strategy selection, as well as screening for chemo- or radio-resistance of tumors.

However, challenges remain concerning the sensitivity, specificity, and applicability of potential circulating miRNA biomarkers and future work will warrant their utility as cancer biomarkers for clinical applications.

## 9. Challenges Involving miRNAs as Biomarkers in Clinical Applications

Because of the aggressive nature of pancreatic cancer and many others, and the lack of reliable and robust biomarkers for early diagnosis, prognosis, and prediction of disease course, miRNAs may provide an additional potential tool to facilitate diagnosis, prediction, monitoring, and management of various therapeutics used in the treatment of pancreatic cancer and many other cancers and potentially improve low survival rates [129].

As discussed in the literature [264], circulating miRNA levels reflect the physiological or pathological status of a person, thus the analysis of miRNA patterns for various disease states may not only lead to novel approaches for the diagnosis and prognosis of disease or condition but also for the selection of appropriate therapies and as predicter biomarkers for monitoring therapy response.

In addition, changes in miRNA signatures provide additional information about the gene networks and molecular pathways that mediate the pathology and regulate the interindividual variability of medical treatments [265].

Furthermore, miRNAs are highly stable, have a long half-life in biological specimens, and the methods for their analysis are readily available and well-established and do not require any special handling and can be applied to samples currently available.

Similarly, profiling of circulating miRNAs can be done with relatively high sensitivity and high specificity, depending on the material available and method used, and with relatively low cost using readily available standard techniques already employed in clinical laboratories such as Northern blot, microarray, quantitative real-time PCR (qRT-PCR), next-generation RNA sequencing, single molecular sequencing, and in situ hybridization [264,266,267].

As discussed in the literature [225], miRNAs have potential in pancreatic cancer management as biomarkers that may help in treatment selection or response evaluation as well as potential therapeutic targets. Despite extensive research on miRNAs as biomarkers in various cancers, data concerning miRNAs’ role in pancreatic cancer are relatively few.

However, despite their potential clinical applications, several challenges including widespread inconsistencies have been observed among the studies which limit the full application of circulating miRNAs as biomarkers in clinical practice [268]. To address these challenges, more research is needed in the detection of miRNAs with high specificity and sensitivity in specimens where RNA material is limited.

In addition, validation of the potential miRNA biomarkers in preclinical patient-derived in vitro and in vivo models, improved bioinformatics to better understand the various pathways of miRNA dysregulation, and validation studies in larger prospective and randomized clinical trials are needed to establish the clinical relevance of miRNAs as biomarkers in pancreatic cancer.

### 9.1. Heterogeneity of Methodologies, miRNA Source Specimens, and Lack of Controls

The main limiting factors in the application of miRNAs as biomarkers in clinical practice include the lack of consensus on methodologies and the use of different biological fluids such as serum, plasma, or extracellular vesicles (EVs), and others for miRNA profiling which introduces biological bias by the use of different matrices [204], there are also methodological and technical biases regarding the distinct isolation protocols.

Other preanalytical and analytical aspects such as technical variation involving RNA isolation, the lack of robust internal controls, and the impact of confounding factors such as sex, age, and concurrent therapies may affect the miRNA levels [269].

### 9.2. Sample Source and Physiological Variations and Low RNA Input

Other challenges involving miRNAs include a low concentration of miRNAs in circulation, sensitivity (a test’s ability to designate an individual with a disease as positive), specificity (a test’s ability to assess the exact component in a mixture), and selectivity (a test’s ability to differentiate the components in a mixture from each other) of the methods used for detecting miRNAs [266], cross-reactivity of miRNAs with different pathologies or disease states and normal physiological states, and stages of a particular disease.

In addition, as discussed in the literature [267], several other factors affect miRNA expression levels, including physiological fluctuations of miRNA expression levels during various cellular processes such as cell division, maturation, differentiation, and development [75,270,271,272,273,274] and age [275,276] or are affected by changes in environmental factors or exposures such as drug treatment and others [277,278].

In addition, miRNAs that are secreted in disease state by specific cells associated with disease, intracellular miRNAs, may also be secreted by other cell types through exosomes and conjugate to RNA-binding proteins, or they may be passively secreted after apoptosis or tissue injury [191].

As for disease pathogenesis, changes in cellular and circulating levels of miRNAs can either be a cause or result of developing pathogenic processes. Also, because miRNA levels have been shown to have a strong correlation with disease progression in various disease types including cancer and many others [21,22,30,31,32,33,34,35,36,37,38,39,40,41,42,43,44,45,46,53,55,56,57,58,60,61,64,66,279,280,281], several of these aspects must be considered when conducting miRNA measurements as disease biomarkers.

Furthermore, there have been variations in results among studies that often result from using patient tissue specimens which contain heterogeneous cell populations (e.g., the ductal, acinar, and islet cells, along with other inflammatory, fibroblastic components that will form the tumor microenvironment). Due to this intratumoral heterogeneity, the prediction of target genes of miRNAs is one of the major challenges of assigning miRNAs in large-scale applications, as each miRNA regulates more than one target and each target might be regulated by multiple miRNAs. In addition, the type of miRNAs might change during the course and stage of cancer, which further complicates the target prediction; however, this can also be beneficial for assigning a specific miRNA or several miRNAs to a specific stage of cancer. Therefore, more innovative approaches for predicting miRNA targets might be required to validate the predicted targets.

Several recent studies have demonstrated that liquid biopsy-based methods can accurately predict tumor response in pancreatic cancer and that sensitivity and specificity are increased when combined with CA19-9 [5,282,283,284,285,286,287,288]; however, there is not enough evidence to demonstrate that liquid biopsy improves patient outcomes or reduces unnecessary chemotherapy.

### 9.3. Methodological and Technical Variations in Detecting miRNAs

One of the main challenges in miRNA biomarker discovery is the poor reproducibility across technologies.

Sensitive and highly specific miRNA detection is required not only for studying physiological processes regulated by these RNAs but also for better understanding of relevant pathologies and their development as potential biomarkers. In addition, research on miRNA biogenesis, localization, function, and dysfunction often needs to be studied in a single cell, with single molecule sensitivity, and spatial as well as temporal resolution [267]. Further, depending on the modality of detection, assay sensitivities remain variable and are impacted by clinical stage, tumor burden, location of primary and metastatic tumor sites, and rates of cell turnover [5]. As a result, the detection of miRNAs can be affected by methods, measurement principles, and other technical factors [130].

There are several detection methods used for miRNA profiling, including Northern blot, microarray, qRT-PCR, next-generation RNA sequencing, single molecular sequencing, and in situ hybridization [266,267], and each of these methods has its strengths and limitations [289].

While assays such as Northern blot, qRT-PCR, microarray, next Generation RNA sequencing, and single molecular sequencing all provide some level of information about the miRNAs profiled, the specificity and sensitivity of each assay, as well as the ability to conduct them in high-throughput format and starting material as well as cost are highly variable. For example, Northern blot has high specificity and middle sensitivity and is an established technique but requires a large amount of sample, is tedious, and has very low throughput. qRT-PCR has moderate specificity and high sensitivity and is an established technique and can be automated but is medium throughput and cannot identify new miRNAs. Microarray has low specificity and low to middle sensitivity with low cost, high-throughput, and is fast; however, microarray is semiquantitative and cannot identify new miRNAs.

In situ hybridization has low specificity and low to middle sensitivity with low cost, low throughput, and slow processing of samples and requires a high level of expertise.

Next-generation RNA sequencing has very high specificity and middle sensitivity with middle high-throughput and whole-genome sequencing capability but is expensive and takes time to process samples and requires a high level of expertise.

Single molecular sequencing has moderate specificity and high sensitivity with middle high-throughput and whole-genome sequencing capability and is relatively fast but is very expensive.

Despite the advances in the development of next-generation RNA sequencing platforms aimed to analyze ultra-low quantities of RNA material, many of these approaches are biased and poorly reproducible with other RNA detection methods [290] due to the low RNA input requiring longer amplification cycles hence resulting in the introduction of amplification biases and the presence of overrepresented sequences with low to no clinical relevance [291].

In addition, the identification and quantification of miRNAs are challenging because of their short length, the existence of miRNA isomers, high homology within miRNA families, and O-methyl 3′ modifications [292].

### 9.4. Variations in Data Analysis and Bioinformatics Tools in the Analysis of miRNAs

Moreover, as discussed recently in the literature [190], several bioinformatic tools, databases, and algorithms have been developed in recent years that are available online to predict regulatory targets and mechanistic functions of each miRNA [101,293].

Importantly, multiple independent miRNA target prediction algorithms such as TargetScan [101] are used concurrently to increase the predictive power of these algorithms and for the prediction of miRNA-binding sites in their protein-coding genes and the biological pathways they regulate.

Furthermore, bioinformatic platforms including IPA/Ingenuity, Kegg, and others are being used to predict biological pathways and disease states that are regulated by miRNAs. Thus, streamlining miRNA research with miRNA bioinformatics platforms is key to facilitating the identification and development of miRNAs as clinically relevant biomarkers.

However, predicted targets and mechanistic functions of candidate miRNA biomarkers by various bioinformatics tools need validation in appropriate in vitro and in vivo preclinical models before testing them in clinical studies. For example, various preclinical models, including various immortalized human cell lines, patient-derived cells, and models such as organoids and in vivo patient-derived xenograft models, have been used to investigate the mechanisms of miRNA candidates.

One challenge is the specificity of the detection method for miRNAs due to their high degree of sequence homology, which is closely related to the type, medication, and prognosis of related diseases [266]. A recent report reviews [266] different methods used in miRNA detection.

Consequently, there is a need for the development of novel miRNA detection and analysis methods, technology platforms, and bioinformatics tools for easy, rapid, accurate, quantitative, robust, reproducible, and cost-effective detection and analysis of miRNAs in different types of specimens with high sensitivity and specificity and this need has attracted some attention from assay developers and biotech researchers [266,294,295,296].

Because each disease is fundamentally the consequence of abnormalities in the entire gene expression network and specific biological pathways, a panel of miRNAs might be a better strategy to understand the disease pathobiology. In many studies, the combination of a panel of miRNAs has been performed to increase the discriminative potential of the biomarker set [37,38,41,42,43,44,190].

Recently, due to its ability to analyze and aid in the interpretation of large datasets, artificial intelligence (AI), including machine learning algorithms, has been adapted for biomarker discovery, and has become a popular alternative statistical approach [264]. The advantages that AI provides are multiple. For example, the AI approach is not solely dependent on datasets involving circulating miRNAs, pharmacological, clinical patient data, and sociodemographic patient data to discriminate disease cases from controls or predict disease risk or outcome but is also capable of combining the value of several genes to produce a miRNA signature with predictive potential [264]. The value of this approach has shown its utility in the evaluation of cardiovascular conditions [297], suggesting that the results produced from these emerging approaches and tools will lead to identification of more robust miRNA biomarkers with stronger discriminative potential.

### 9.5. Poor Experimental Design in Clinical Studies

To date, several registered human clinical studies in the “clinicaltrials.gov” (accessed on 6 July 2023) database are exploring the utility of miRNAs as biomarkers across human conditions including various types of cancers, diabetes, lupus, coronary heart disease, epilepsy, depressive disorder, stroke, Addison’s disease, influenza, liver disease, and even toxic exposure to agents such as acetaminophen and several phase four trials that monitored select miRNAs as biomarkers for disease progression in patients receiving FDA-approved drugs [190].

In addition, as illustrated in Table 2, there are several ongoing or completed prospective human clinical trials for the assessment of the clinical utility of circulating miRNAs in pancreatic cancer including miRNA-25 as a diagnostic tool in pancreatic cancer (NCT03432624).

However, there are several challenges associated with these studies, including the insufficient sensitivity, specificity, and selectivity of miRNAs, especially in early-stage disease, due to the low abundance of circulating miRNAs, variability of detection methods used to measure miRNAs, and the lack of standardization of miRNA isolation and detection methods. The detection of miRNAs can be influenced by methods and measurement principles and other technical factors [130]. More importantly, there is a lack of comprehensive validation of miRNA signatures for a disease of interest in large prospective randomized controlled human trials, which is key to further validating the sensitivity, specificity, selectivity, and applicability of the candidate disease-specific miRNAs as biomarkers.

Notably, many miRNAs reported in the literature as associated with cancer and other conditions may be the outcome of underpowered studies due to small sample sizes, which often results in poor replication of results between studies and limits the potential utility of miRNAs as biomarkers in clinical applications [298].

Furthermore, due to many confounding factors used as criteria in clinical studies, clinical trials must consider avoiding the hurdle of limited sample size [23].

For example, although case-control studies based on patients with different pathologies and healthy individuals have been used in discovery studies to serve as controls and are useful for molecular phenotyping, this approach often does not sufficiently consider confounding factors such as age, sex, ethnicity, lifestyle, treatment history, history of diseases, concurrent therapies and treatments and comorbidities [264].

In addition, the high cost of analysis of miRNAs in a large number of samples, lack of replication of the results in independent clinical trials, and lack or low use of prospective, adequately powered large, and well-controlled randomized trials are the other factors that limit the translation and clinical applicability of the experimental findings.

Until these issues are addressed and the data published, widespread use of circulating miRNAs as biomarkers for diagnosis, monitoring of treatment response, and detection of recurrence of pancreatic cancer is unsupported.

## 10. Conclusions and Future Perspectives

Because many miRNAs are abnormally expressed or mutated in most malignant cancers such as pancreatic cancer and others, and since miRNAs could act as an oncogene or a tumor suppressor gene, they have emerged as potential biomarkers as well as therapeutic targets. In particular, circulating miRNAs which can be detected in various body fluids have become promising effective non-invasive cancer biomarkers for clinical application. Toward this goal, in recent years, significant progress has been made in miRNA profiling in liquid biopsy specimens. These non- or minimally invasive platforms thus hold significant promise and are likely to soon improve screening, diagnosis, cancer subtype classification, prognosis, and monitoring of therapy response and surveillance of pancreatic cancer and many other cancers and malignancies. The wide adoption of next-generation molecular profiling technologies such as high-throughput next-generation RNA sequencing including single-cell RNA sequencing is providing researchers with better insights into gene dysregulation in tumorigenesis and the discovery of novel RNA biomarkers including miRNAs and other non-coding-RNAs for cancer screening as well as for developing RNA-targeted or RNA-based cancer therapies.

Moreover, in addition to tissue-specific miRNA profiling, circulating miRNAs are being explored as potential minimally invasive biomarkers for clinical application, including cancer screening in the early stage, subtype classification, and predicting drug sensitivity for treatment strategy selection, as well as screening the chemo- or radio-resistance of tumors. Furthermore, miRNAs may provide important insights into tumor evolution and mechanisms of therapy resistance.

However, there are many challenges concerning the applicability of circulating miRNAs as biomarkers. For example, widespread adoption of miRNA biomarkers in routine clinical practice has been hindered by insufficient sensitivity, specificity, and selectivity of miRNAs, especially in early-stage disease and across different types of diseases. Because of this, a comprehensive validation and standardization of current miRNA profiling and analysis methodologies, as well as confirmation of candidate circulating miRNAs in large prospective randomized controlled trials to further validate the sensitivity, specificity, selectivity, and applicability of potential circulating miRNA biomarkers, are required.

It is also worth mentioning that, due to many confounding factors used as criteria in clinical applications such as age, sex, ethnicity, lifestyle, treatment history, history of disease, and others, it is mandatory to avoid the obstacle of limited sample size.

Given the potential to change clinical practice, miRNA biomarkers, especially those that can be used in early diagnosis of aggressive cancers such as pancreatic cancer, are rapidly improving and are expected to become part of the standard toolkit in pancreatic cancer management.

However, there needs to be significant efforts put in place to accomplish the clinical application of miRNAs as biomarkers, in particular, efforts need to focus on overcoming current methodological, technical, or analytical challenges and harmonization of miRNA isolation and quantification methods alongside the use of standard operating procedures, the development of automated and standardized assays, and the miniaturization of the methods to improve reproducibility between independent studies and more importantly for bench-to-bedside translation of the miRNA biomarkers for clinical applications.

To facilitate these, there is a need for the establishment of specific guidelines and protocols on best practices to augment the quality of the miRNA biomarker studies.

Furthermore, the implementation of miRNAs as biomarkers in clinical research also requires biobanks accessible to researchers to address the low availability of large and independent cohorts and importantly, cost-effectiveness.

Furthermore, a systematic and collaborative strategy that incorporates basic and clinical research and associated investigators, industry partners, and governmental agencies is key to help in translating early miRNA biomarkers into clinical applications.

Collectively, key to the success lies in the establishment of a research ecosystem favoring and supporting collaboration across various stakeholders from basic to clinical research to industry partners and improvement of funding opportunities provided by governmental agencies to facilitate the translation from bench to bedside.

## Figures and Tables

**Figure 1 ijms-24-13340-f001:**
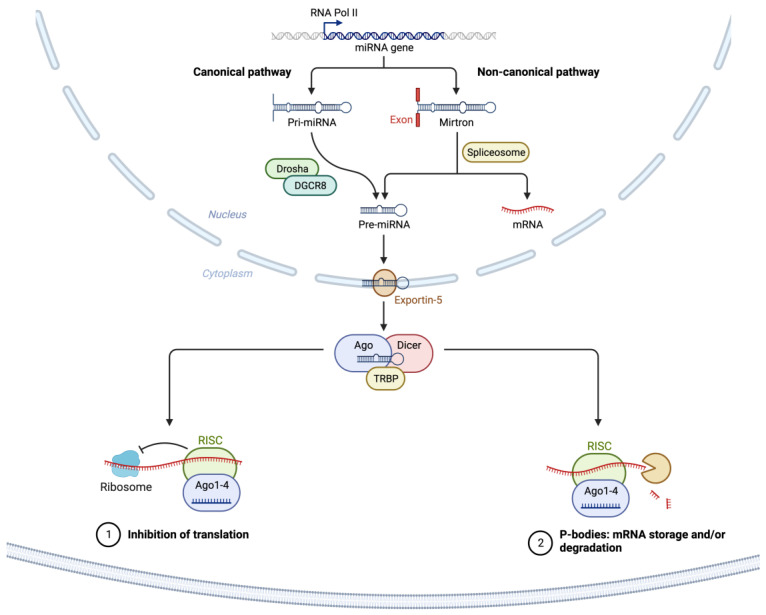
Schematic of canonical and noncanonical miRNA biogenesis, processing, and target RNA translational suppression or degradation. The canonical miRNA pathway generates pri-miRNAs from transcripts by miRNA genes encoded in intronic, exonic, or intergenic regions. The pri-miRNA transcripts are then processed by Drosha/DGCR8 into pre-miRNAs. Intronic pre-miRNAs of the noncanonical mirtron pathway are formed by splicing, debranching, and trimming short introns without the involvement of Drosha processing. The pre-miRNA transcripts produced by both the canonical and noncanonical pathways are then exported from the nucleus to the cytoplasm by Exportin 5. This is followed by subsequent Dicer cleavage within the RISC loading complex (RLC) and subsequent unwinding of the miRNA/miRNA duplex via Argonaute, and TRBP-dependent loading into the RNA-induced silencing complex (RISC). The binding of target mRNAs to miRNAs in RISC is followed by suppression of translation and/or mRNA degradation within p-bodies in the cytosol. Created with BioRender.com (accessed on 6 July 2023). Abbreviations: AGO2: Argonaute RISC Catalytic Component 2; DICER1: Dicer 1, Ribonuclease III; EGFR: Epidermal growth factor receptor 2; DGCR8: DiGeorge Syndrome Critical Region 8; DROSHA: Drosha Ribonuclease III; mRNA: messenger RNA; miRNA: micro RNA; P-bodies: Processing bodies; pre-miRNA: precursor microRNA; pri-miRNA: primary microRNA; RNA Pol II: RNA polymerase II; TRBP: AR RNA-binding protein; XPO5: Exportin 5.

**Figure 2 ijms-24-13340-f002:**
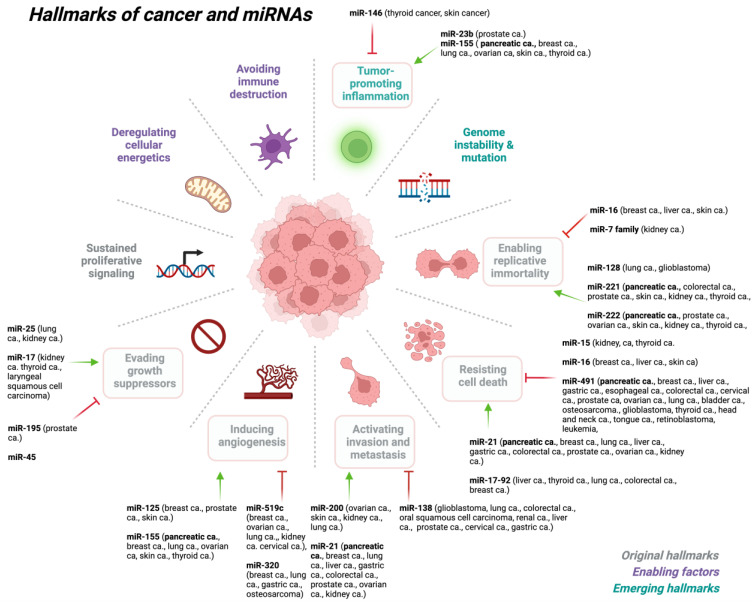
miRNAs play key roles at various stages of cancer development. Created with BioRender.com (accessed on 6 July 2023). Abbreviations: ca: cancer.

**Figure 3 ijms-24-13340-f003:**
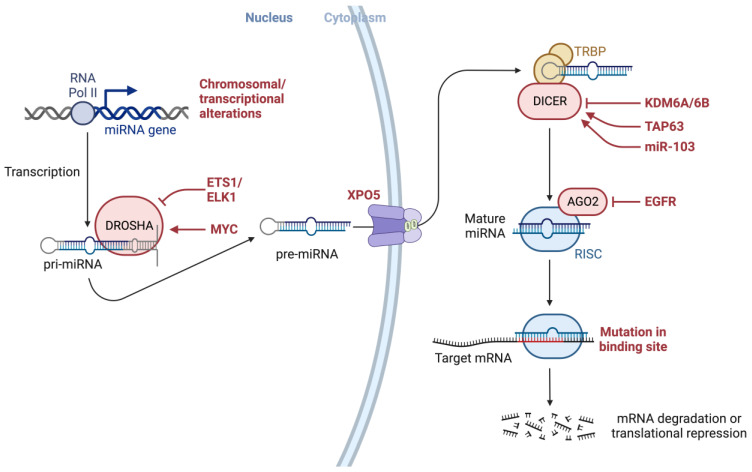
Alterations throughout miRNA biogenesis can affect the availability of target mRNA, including many mRNAs associated with cancer development. The figure also depicts several modulators, proteins such as EGFR, and transcription factors such as ETS1/ELK1, MYC, epigenetic regulators such as demethylating protein KDM6A, a known tumor suppressor, the tumor suppressor gene TAP63, and miRNAs such as miR-103 that may interact with modulators of miRNA-processing machinery. Both KRAS and EGFR are essential mediators of pancreatic cancer development and interact with AGO2 to perturb its function. Created with BioRender.com (accessed on 6 July 2023). Abbreviations: AGO2: Argonaute RISC Catalytic Component 2; DICER1: Dicer 1, Ribonuclease III; DROSHA: Drosha Ribonuclease III; EGFR: Epidermal growth factor receptor 2; EGFR: Epidermal growth factor receptor, ELK1: ETS Like-1 protein Elk-1; ETS1: ETS Proto-Oncogene 1; KDM6A: Lysine (K)-specific demethylase 6A; miRNA: micro RNA; mRNA: messenger RNA; MYC: MYC Proto-Oncogene; pre-miRNA: precursor microRNA; pri-miRNA: primary microRNA; RNA Pol II: RNA polymerase II; TAP63: Tumor protein p63; XPO5: Exportin 5.

**Figure 4 ijms-24-13340-f004:**
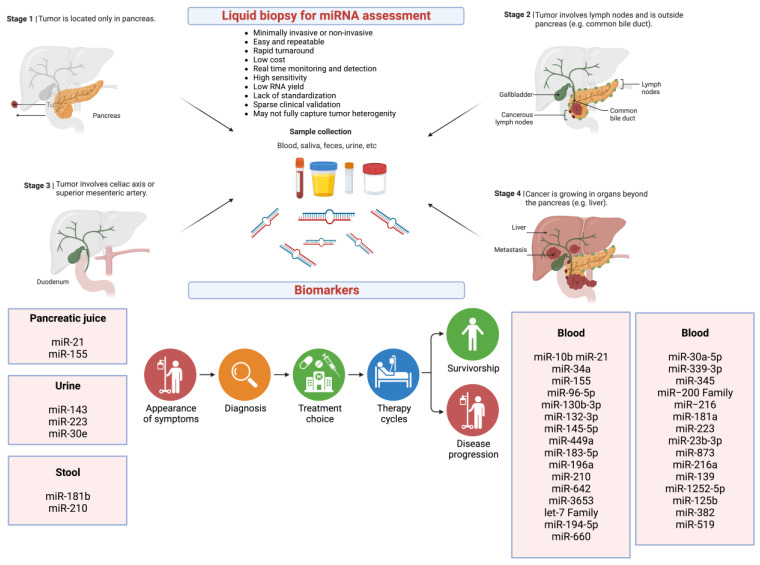
miRNAs as potential liquid biopsy biomarkers for pancreatic cancer. Created with BioRender.com (accessed on 6 July 2023). Abbreviations: Let-7 family: lethal-7 family of 12 miRNAs; miRNA: micro RNA; miR: micro RNA.

**Table 1 ijms-24-13340-t001:** Recent clinical findings on miRNAs as prognostic markers and possible mechanisms of action and survival status in pancreatic cancer. Abbreviations: Akt: Ak strain transforming; Anxa3: Annexin A3; ATRX: Alpha-thalassemia/mental retardation, X-linked; BCL2: B-cell lymphoma 2; CDK6: Cell division protein kinase 6; E-cadherin: Epithelial cadherin; KRAS: Kirsten rat sarcoma virus; FOXO1: Forkhead box protein O1; HIV-1 Tat interactive protein 2 (HTATIP2); HRAS: Harvey rat sarcoma virus; IL-6R: Interleukin 6 receptor; TIP30: JAK: Janus kinase; miR: microRNA; mRNA: messenger RNA; NA: not applicable; NEDD9: Neural precursor cell expressed developmentally down-regulated protein 9; NOTCH: Neurogenic locus notch homolog protein; PD-L1: Programmed death-ligand 1; PDCD4: Programmed cell death protein 4; PI3K: Phosphoinositide 3-kinase; PLEK2: Pleckstrin 2; PTEN: Phosphatase and tensin homolog; RalB: RAS like proto-oncogene B; RKIP: RAF-kinase inhibitor protein; ROCK1: Rho-associated coiled-coil kinases; TP53INP: Tumor protein p53-inducible nuclear protein; TRIM71: Tripartite motif containing 71: WT1: Wilms’ tumor suppressor gene1; ZEB1: Zinc finger E-box-binding homeobox 1; ZIC1: Zinc finger of the cerebellum.

miRNA	Expression Status	Type of Cancer	Function	Clinical Implications	References
miR-10b	Upregulated	Pancreatic cancer	TIP30—mRNA inhibition	Poor Survival	[227,228]
miR-21	Upregulated	Pancreatic cancer Pancreatic neuroendocrine tumors	PTEN, PDCD4, IL-6R, CDK6 —mRNA inhibition Metastatic disease Proliferative activity	Worse Survival Tumor grade	[229,230,231,232,233]
miR-34a	Upregulated	Pancreatic cancer	NOTCH, BCL2, CDK6—mRNA inhibition Somatostatin, gastrin, and serotonin expression	Better Survival	[209,234,235,236]
miR-155	Upregulated	Pancreatic cancer	TP53INP—mRNA inhibition	Poor Survival	[237,238]
miR-96-5p	Upregulated	Pancreatic neuroendocrine tumors	Oncogenic, FOXO1a inhibition	High tumor grade	[239]
miR-130b-3p	Upregulated	Pancreatic neuroendocrine tumors	NA	High tumor grade	[239]
miR-132-3p	Upregulated	Pancreatic neuroendocrine tumors	Tumor-suppressing and tumor-promoting function	Low tumor grade Vascular invasion Somatostatin expression	[236]
miR-145-5p	Upregulated	Pancreatic neuroendocrine tumors		High tumor grade Lymphatic invasion Serotonin expression	[236]
miR-449a	Upregulated	Pancreatic neuroendocrine tumors	Oncogenic via histone deacetylases 3/4	High tumor grade, mitotic and proliferative activity, lymph-node invasion	[236]
miR-183-5p	Upregulated	Pancreatic neuroendocrine tumors	Tumor suppressor	High tumor grade Tumor size Somatostatin-receptor expression	[236]
miR-196a	Upregulated	Pancreatic neuroendocrine tumors	NA	Advanced tumor Lymph-node invasion High mitotic and proliferative activity Recurrence	[240]
miR-210	Upregulated	Pancreatic neuroendocrine tumors	Oncogenic	Metastatic disease	[232,241]
miR-642	Upregulated	Pancreatic neuroendocrine tumors	Oncogenic	Proliferative activity	[241]
miR-3653	Upregulated	Pancreatic neuroendocrine tumors	Oncogenic associated with ATRX mutations	Metastatic disease	[242]
let-7 Family	Downregulated	Pancreatic cancer	KRAS, HRAS, TRIM71	Poor Survival	[243,244]
miR-194-5p	Downregulated	Pancreatic neuroendocrine tumors	NA	High tumor grade	[239]
miR-660	Downregulated	Pancreatic neuroendocrine tumors	NA	Metastatic disease	[232]
miR-30a-5p	Downregulated	Pancreatic neuroendocrine tumors	NA	Metastatic disease	[245]
miR-339-3p	Downregulated	Pancreatic neuroendocrine tumors	NA	Metastatic disease	[232]
miR-345	Downregulated	Pancreatic neuroendocrine tumors	NA	Metastatic disease	[232]
miR−200 Family	Downregulated	Pancreatic cancer	E-cadherin, ZEB1	Better Survival	[234,246,247]
miR−216	Downregulated	Pancreatic cancer	ROCK1	Poor Survival	[248,249]
miR-181a	Upregulated	Pancreatic cancer	RKIP—mRNA inhibition and induces epithelial–mesenchymal transition like cancer stem cell phenotype	Poor Survival	[250]
miR-223	Upregulated	Pancreatic cancer	ZIC1—mRNA inhibition via PI3K/Akt/mTOR signaling pathway	Poor Survival	[251]
miR-23b-3p	Upregulated	Pancreatic cancer	PTEN—mRNA inhibition via the JAK/PI3K and Akt/NF-kappa B signaling pathways	Poor Survival	[252]
miR-873	Downregulated	Pancreatic cancer	PLEK2—mRNA inhibition via pleckstrin-2-dependent PI3K/AKT pathway	Better Survival	[253]
miR-216a	Downregulated	Pancreatic cancer	WT1—mRNA inhibition and regulates KRT7 transcription	Poor Survival	[254]
miR-139	Downregulated	Pancreatic cancer	RalB—mRNA inhibition via the Ral/RAC/PI3K pathway	Poor Survival	[255]
miR-1252-5p	Downregulated	Pancreatic cancer	NEDD9 mRNA inhibition. MicroRNA-1252-5p, regulated by Myb regulates NEDD9 mRNA inhibition	Poor Survival	[256]
miR-125b	Downregulated	Pancreatic cancer	NEDD9—mRNA inhibition via PI3K/AKT signaling pathway	Poor Survival	[257]
miR-382	Downregulated	Pancreatic cancer	Anxa3—mRNA inhibition via PI3K/Akt signaling pathway	Poor Survival	[258]
miR-519	Downregulated	Pancreatic cancer	PD-L1—mRNA inhibition	Poor Survival	[259]

**Table 2 ijms-24-13340-t002:** Current miRNA human clinical trials as diagnostic biomarkers. Source: Clinicaltrials.gov (accessed on 6 July 2023). Key words: miRNA and pancreatic cancer. Data acquisition: 30 June 2023. Abbreviations: cfDNA: cell-free DNA; MEN1: Multiple endocrine neoplasia type 1; Nab-Paclitaxel: nanoparticle albumin–bound paclitaxel.

Study Title	Conditions	Interventions	Study Type	Phase	Status	NCT Number
Detection of MicroRNA-25 in the Diagnosis of Pancreatic Cancer	Carcinoma, Pancreatic Ductal	Diagnostic Test: Serum MicroRNA-25 detection	Observational	NA	Unknown	NCT03432624
The Role of MicroRNA in the Diagnosis, Prognosis and Response to Treatment in Pancreatic Cancer	Pancreatic Cancer Stage III Pancreatic Cancer Stage IV Pancreatic Ductal Adenocarcinoma Pancreatic Neoplasms	Procedure: Blood draw	Observational	NA	Recruiting	NCT04406831
Identify microRNAs in Cachexia in Pancreatic Carcinoma	Resectable Pancreatic Adenocarcinoma	Other: Pancreatic cancer microRNA and messenger RNA expression	Observational	NA	Recruiting	NCT05275075
Lipidomics, Proteomics, Micro RNAs, and Volatile Organic Compounds (VOC)	Pancreatic Neoplasms	Other: blood and bile	Observational	NA	Active, not recruiting	NCT02531607
U01-Biomarkers for Noninvasive and Early Detection of Pancreatic Cancer	Pancreatic Cancer		Observational	NA	Recruiting	NCT03886571
iDentification and vAlidation Model of Liquid biopsY Based cfDNA Methylation and pRotEin biomArKers for Pancreatic Cancer (DAYBREAK Study)	Cancer		Observational	NA	Recruiting	NCT05495685
Radiofrequency Ablation Combined With S-1 for Pancreatic Cancer with Liver Metastasis	Carcinoma, Pancreatic	Procedure: radiofrequency ablation Drug: S-1	Interventional	Phase 2	Unknown	NCT02634502
BIOmarkers in Patients with Pancreatic Cancer (“BIOPAC”)	Pancreatic Cancer		Observational	NA	Recruiting	NCT03311776
AssesSment of Early-deteCtion basEd oN liquiD Biopsy in PANCEATIC Cancer (ASCEND-PANCREATIC)	Cancer		Observational	NA	Recruiting	NCT05556603
Validation of Useful Markers Generated by Next Generation Bio-data Based Genome Research and Cohort Study	BCL2 Gene mRNA Overexpression		Observational	NA	Completed	NCT02807896
Project CADENCE (CAncer Detected Early caN be CurEd)	Pancreatic Cancer Thoracic Cancer Ovarian Cancer Liver Cancer Prostate Cancer Gastric Cancer Colorectal Cancer Breast Cancer Esophageal Cancer		Observational	NA	Recruiting	NCT05633342
Metabolomics and Genetic Diagnosing Pancreatic Neuroendocrine Tumors in MEN1 Patients	Multiple Endocrine Neoplasia		Observational	NA	Recruiting	NCT03048266
Gemcitabine, Nab-Paclitaxel, and Bosentan for the Treatment of Unresectable Pancreatic Cancer	Stage III Pancreatic Cancer AJCC v8	Drug: Bosentan	Interventional	Phase 1	Recruiting	NCT04158635
Atu027 Plus Gemcitabine in Advanced or Metastatic Pancreatic Cancer (Atu027-I-02)	Carcinoma, Pancreatic Ductal	Drug: Atu027 & gemcitabine in lead in safety period Drug: Atu027 & gemcitabine in treatment arm 1	Interventional	Phase 1	Completed	NCT01808638
A Prospective Translational Tissue Collection Study in Early and Advanced Pancreatic Ductal Adenocarcinoma and Pancreatic Neuroendocrine Tumours to Enable Further Disease Characterisation and the Development of Potential Predictive and Prognostic Biomarkers	Pancreatic Adenocarcinoma		Observational		Unknown	NCT03840460

## Data Availability

The data presented in this study are openly available in https://seer.cancer.gov, https://pubmed.ncbi.nlm.nih.gov/, https://clinicaltrials.gov/, and https://www.genomicseducation.hee.nhs.uk/genotes/knowledge-hub/non-coding-dna, all accessed on 26 July 2023.

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
