# Peer review of "Circulating microRNAs as Potential Biomarkers in Pancreatic Cancer—Advances and Challenges"

_ijms, 2023, doi:10.3390/ijms241713340_

Round 1

Reviewer 1 Report

It is a very interesting review article that describes the potential role of miRNAs in pancreatic cancer. The article is well written but shows very little data about the advances and challenges of miRNA biomarkers. miRNAs in cancer are already well-established field rather than describing miRNAs in cancer review needs to concentrate on the challenges facing in implementing it as a biomarker.

Author Response

I thank the reviewer for the helpful comments which were very helpful in revising the manuscript to convey the message more clearly. Below is a point-by-point response to each comment.

Response to reviewer 1: The author thanks to the reviewer for the important critique and helpful comments. The manuscript has been revised accordingly and it now contains more details about the challenges associated with the detection and implementation of miRNAs as biomarkers in the “Challenges involving miRNAs as biomarkers in clinical applications” section. All revised text is in green or red or purple font.

Reviewer 2 Report

The topic of the paper is quite relevant as pancreatic cancer is often deadly and any measure that allows its early diagnostics is very welcome. However, several other papers already mention the role of circulating miRNAs (and RNAs) as biomarkers in pancreatic cancer. Can  you relate your paper with the previous published ones and justify the relevance of yours relative to others, what does this paper brin of novelty and what are its strengths relative to already available information? 

Did the author ask permission for the adaptation of figure 2 used in the paper? It Is very similar to the original one. Could the author update the figure including miRNAs that are not in the original referenced paper and also in which types of cancers they are involved, where those microRNAs are expressed?

Again, figure 4 is also very similar to the original one. Could the author provide selective information for this figure based only on pancreatic cancer, which is the topic the current paper? The same applies to the information of table 1. If no information is available or if is very limited maybe you could just add it as text instead of making a figure and reference paper 18 to invite readers for a full paper of the topic in other tissues. You can also mention other types of cancer in the paper but pancreatic cancer should be your “reference” cancer. 

In lines 274 and 275 the word because is used twice in the same sentence. 

Although it is not the topic of your review, can  you explore a little further the discussion of the use of miRNAs in clinical trials, major advantages and disadvantages, based on results from table  3?

The english is quite good. Just found some repeating words in some sentences such as the one mentioned above: In lines 274 and 275 the word because is used twice in the same sentence. 

We advise the author to revise the paper for other potential mistakes. 

Author Response

Response to reviewer 2: I thank the reviewer for the helpful comments which were very helpful in revising the manuscript to convey the message more clearly. Below is a point-by-point response to each comment  (in red). 

Comments and Suggestions for Authors

The topic of the paper is quite relevant as pancreatic cancer is often deadly and any measure that allows its early diagnostics is very welcome. However, several other papers already mention the role of circulating miRNAs (and RNAs) as biomarkers in pancreatic cancer.

Can you relate your paper with the previous published ones and justify the relevance of yours relative to others, what does this paper bring of novelty and what are its strengths relative to already available information?

Response to reviewer 2: Author thanks for the important points the reviewer has raised.

Current paper incorporates recent findings on circulating miRNAs involving pancreatic cancer as well as  miRNAs that are currently in clinical trials (Data acquisition: June 30, 2023) as potential biomarkers in pancreatic cancer.

In addition, a detailed analysis for the challenges involving the detection, detection methodologies, variation in source specimens and lack of controls, sample source and physiological variations and low RNA input, methodological and technical variations in detecting miRNAs, variations in data analysis and bioinformatics tools in the analysis of miRNAs, poor experimental design in clinical studies and other technical challenges such as issues regarding reproducibility of data, challenges involving data analysis and interpretation, including limited or lack of validation of candidate miRNA signatures in large prospective randomized controlled human trials have been discussed.

Did the author ask permission for the adaptation of figure 2 used in the paper? It Is very similar to the original one. Could the author update the figure including miRNAs that are not in the original referenced paper and also in which types of cancers they are involved, where those microRNAs are expressed?

Response to reviewer 2: The Figure 2 has been updated significantly such that the miRNAs associated with specific hallmarks of cancer now have information about specific cancer types/histologies that these miRNAs are involved. This information is not present in the original referenced paper.

Again, figure 4 is also very similar to the original one. Could the author provide selective information for this figure based only on pancreatic cancer, which is the topic the current paper?

Response to reviewer 2: Figure 4 and Table 2 have been extensively revised which now includes new data with relevant references.

The same applies to the information of table 1. If no information is available or if is very limited maybe you could just add it as text instead of making a figure and reference paper 18 to invite readers for a full paper of the topic in other tissues. You can also mention other types of cancer in the paper but pancreatic cancer should be your “reference” cancer.

Response to reviewer 2: Table 1 and 4 have been deleted and only mentioned within the text  with appropriate citation to the reference.

In lines 274 and 275 the word because is used twice in the same sentence.

Response to reviewer 2: the duplication of the word “because” in lines 274 and 275 has been corrected.

Although it is not the topic of your review, can you explore a little further the discussion of the use of miRNAs in clinical trials, major advantages and disadvantages, based on results from table 3?

Response to reviewer 2: Several additional points on the use of miRNAs as biomarkers in basic and clinical research including clinical trials have been discussed in "the “Challenges involving miRNAs as biomarkers in clinical applications” section. The new text is indicated in red font.

Comments on the Quality of English Language

The English is quite good. Just found some repeating words in some sentences such as the one mentioned above: In lines 274 and 275 the word because is used twice in the same sentence.

Response to reviewer 2: The duplication of the word “because” in lines 274 and 275 has been corrected.

We advise the author to revise the paper for other potential mistakes.

Response to reviewer 2: The paper has been revised for other potential errors and new text is indicated in green, red, and purple.

Submission Date

05 July 2023

Date of this review

25 Jul 2023 09:28:35

Reviewer 3 Report

Author aimed to summarize results in circulating miRNAs with potential for pancreatic cancer biomarkers.

This manuscript has unclear structure; it is highly un-balanced and full of general phrases. Many facts are repeated several times, also by almost the same sentences. In most of text, author focused on miRNAs in any types of tumors, which could be only introduction to the results from pancreatic cancer studies.

·         Introduction need present the main message of paper, not generally known facts about miRNAs.

·         Figures and Tables: Author used figures and tables from other authors and formulation “Adapted from (ref.)”, but it is usually used, when the new facts or details are added. Figure 2. and 4. were only re-draw by BioRender.com and Tables 1 and 4. are identical with origin papers.

·         Are there any miRNAs associated with pancreatic cancer development (own picture instead Figure 2.)?

·         Row 130: I do not understand why author described the specific features of brain tumor when this manuscript is about pancreatic cancer.

·         Location or missing of references: Rows 55 – 60: There is still in debate number of protein-coding genes. Include the relevant references. Several/many references were inserted only at the end of paragraphs, others at “unusual “ places (in the text, figure legends and conclusions).

·         I did not find similar figure at reference 115 as your Figure 5 (Adapted from: [115]).

Formal errors

·         Re-arrange the list of references according to their row, when they were firstly used in the text including figures and tables.

·         Include the abbreviations after the Table 3.

·         Re-arrange the row of Tables 1 and 2. “

Conclusion:

This manuscript need to be completely re-written; therefore, I recommend rejecting it for publication at IJMS.

Moderate editing.

Author Response

Response to reviewer 3: I thank the reviewer for the helpful comments which were very helpful in revising the manuscript to convey the message more clearly. Below is a point-by-point response to each comment.

Comments and Suggestions for Authors

Author aimed to summarize results in circulating miRNAs with potential for pancreatic cancer biomarkers.

This manuscript has unclear structure; it is highly un-balanced and full of general phrases. Many facts are repeated several times, also by almost the same sentences.

In most of text, author focused on miRNAs in any types of tumors, which could be only introduction to the results from pancreatic cancer studies.

  • Introduction need present the main message of paper, not generally known facts about miRNAs.

Response to reviewer 3: I thank the reviewer for the helpful comments. Introduction has been revised accordingly.

  • Figures and Tables: Author used figures and tables from other authors and formulation “Adapted from (ref.)”, but it is usually used, when the new facts or details are added. Figure 2. and 4. were only re-draw by BioRender.com and Tables 1 and 4. are identical with origin papers.

Response to reviewer 3: I thank the reviewer for the helpful comments. Figure 2 has been revised significantly which now contains new details and data and is not a mere redrawing of the figure from the referenced paper. On the other hand, Figure 4 has been deleted.

Response to reviewer 3: Table 1 and 4 have been deleted and only mentioned within the text  with appropriate citation to the references.

  • Are there any miRNAs associated with pancreatic cancer development (own picture instead Figure 2.)?

Response to reviewer 3: Figure 2 has been updated significantly such that the miRNAs associated with specific hallmarks of cancer now also have information about specific cancer types including pancreatic cancer that these miRNAs are involved. This information is not present in the original referenced paper.

In addition, Figure 4 now includes examples of several miRNAs as potential liquid biopsy biomarkers for pancreatic cancer.

  • Row 130: I do not understand why author described the specific features of brain tumor when this manuscript is about pancreatic cancer.

Response to reviewer 3: Line 130 has been deleted in Figure 1 legend.

  • Location or missing of references: Rows 55 – 60: There is still in debate number of protein-coding genes. Include the relevant references.

Response to reviewer 3: Relevant references regarding the protein-coding genes has been included.

Several/many references were inserted only at the end of paragraphs, others at “unusual “ places (in the text, figure legends and conclusions).

Response to reviewer 3: References in Conclusion and Figure legends have been deleted and inserted in the appropriate places in the main text.

  • I did not find similar figure at reference 115 as your Figure 5 (Adapted from: [115]).

Response to reviewer 3: Figure 4 (previously 5) has been significantly revised which now contains  new details and data as such the original figure in the following reference no longer has much resemblance other than just the general concept.

https://bmccancer.biomedcentral.com/articles/10.1186/s12885-019-6284-y

Daoud, A. Z.; Mulholland, E. J.; Cole, G.; McCarthy, H. O., MicroRNAs in Pancreatic Cancer: biomarkers, prognostic, and therapeutic modulators. BMC Cancer 2019, 19, (1), 1130.

Formal errors

  • Re-arrange the list of references according to their row, when they were firstly used in the text including figures and tables.
  • Include the abbreviations after the Table 3.

Response to reviewer 3: Abbreviations have been included after Figures 1-3 and Table 1-2.

  • Re-arrange the row of Tables 1 and 2. “

Response to reviewer 3: Table 1 has been been deleted only mentioned within the text  with appropriate citation to the reference.

Table 2 has been revised and new data have been included with relevant references.

Conclusion:

This manuscript need to be completely re-written; therefore, Irecommend rejecting it for publication at IJMS.

Comments on the Quality of

Moderate editing.

EnglishLanguage

SubmissionDate

05 July 2023

Date of this review

07 Aug 2023 11:32:51

Round 2

Reviewer 2 Report

After the extensive review by the author, the manuscript quality has improved significantly and all questions posed have been addressed. Thus, I consider it can be published in the current form.

Reviewer 3 Report

Author made a progress in this manuscript and re-wrote it according the recommendations; however, it need an additional revision. 

·         This text could be more readable, if author avoid the general phrases and repeating of the information. Focus only on cancer –> pancreatic cancer, not on all human diseases (or change the title: pancreatic cancer or human diseases???).

For example, first three paragraphs at page 5 could be one, repeating at the page 6 and second paragraph…etc.

·         Instead many references to one general info, insert the results of studies in the context of topic. Select relevant references more carefully. Thereby, some references seem to be redundant.  

For example:

As such, miRNAs and their altered expression have also been recognized as an additional molecular mechanism responsible for the pathological processes of many diseases [25-36] including many cancers [37, 38].  There are also cancers between ref. 25-36!

…. dysregulation of miRNA expression is closely associated with cancer initiation, progression, and metastasis and shown to be associated with the origin, progression, therapeutic response, and patient survival of the disease [59, 61, 62, 96-99].

Abstract: Profiling dysregulated or deregulated miRNAs ?? Is there any difference between them?

Correct typos and English.

Conclusion:

Author need to re-write the manuscript to high informative and well-arranged text with clear structure. I recommend major revision.

Moderate revision.

Author Response

Reviewer 3

Response to reviewer 3: I thank the reviewer for the helpful comments which were very helpful in revising the manuscript to convey the message more clearly. Below is a point-by-point response to each comment.

Comments and Suggestions for Authors – Reviewer 3, Round 2

Author made a progress in this manuscript and re-wrote it according to the recommendations; however, it need an additional revision.

  • This text could be more readable if author avoid the general phrases and repeating of the information. Focus only on cancer –>pancreatic cancer, not on all human diseases (or change the title: pancreatic cancer or human diseases???).

For example, first three paragraphs at page 5 could be one, repeating at the page 6 and second paragraph…etc.

  • Instead, many references to one general info, insert the results of studies in the context of topic. Select relevant references more carefully. Thereby, some references seem to be redundant.

Response to reviewer 3: References citing to a specific information have been inserted in the proper places within the context of the topic rather than at the end of the sentence or paragraph.

For example:

As such, miRNAs and their altered expression have also been recognized as an additional molecular mechanism responsible for the pathological processes of many diseases [25-36] including many cancers [37, 38]. There are also cancers between ref. 25-36!

…. dysregulation of miRNA expression is closely associated with cancer initiation, progression, and metastasis and shown to be associated with the origin, progression, therapeutic response, and patient survival of the disease [59, 61, 62, 96-99].

Response to reviewer 3: Introduction has been revised accordingly and aforementioned first three paragraphs at page 5 and 6 have been consolidated and repeated text has been deleted.

Abstract: Profiling dysregulated or deregulated miRNAs ?? Is there any difference between them?

Response to reviewer 3: Deregulated has been removed from the abstract and elsewhere in the main text and only and now is referred to as “dysregulated” throughout the text.

Correct typos and English.

Response to reviewer 3: The manuscript has been thoroughly checked for potential typos and errors in English.

Conclusion:

Author needs to re-write the manuscript to high informative and well-arranged text with clear structure. I recommend major revision.

Response to reviewer 3: The manuscript has been thoroughly checked for potential typos and errors in English.

Comments on the Quality of English Language

Moderate revision.

Submission Date

05 July 2023

Date of this review

17 Aug 2023 14:33:46
